# Comparative Genomic Analyses of New and Old World Viscerotropic Leishmanine Parasites: Further Insights into the Origins of Visceral Leishmaniasis Agents

**DOI:** 10.3390/microorganisms11010025

**Published:** 2022-12-21

**Authors:** Fernando Tobias Silveira, Edivaldo Costa Sousa Junior, Rodrigo Vellasco Duarte Silvestre, Thiago Vasconcelos dos Santos, Wilfredo Sosa-Ochoa, Concepción Zúniga Valeriano, Patrícia Karla Santos Ramos, Samir Mansour Moraes Casseb, Luciana Vieira do Rêgo Lima, Marliane Batista Campos, Vania Lucia da Matta, Claudia Maria Gomes, Gabriela V. Araujo Flores, Carmen M. Sandoval Pacheco, Carlos Eduardo Corbett, Márcia Dalastra Laurenti

**Affiliations:** 1Parasitology Department, Evandro Chagas Institute, Ananindeua 67030-000, Pará State, Brazil; 2Tropical Medicine Nucleus, Federal University of Pará, Belém 66055-240, Pará State, Brazil; 3Virology Department, Evandro Chagas Institute, Ananindeua 67030-000, Pará State, Brazil; 4Microbiology School, National Autonomous University of Honduras, Tegucigalpa 11101, Honduras; 5Health Surveillance Department, School Hospital, Autonomous University of Honduras, Tegucigalpa 11101, Honduras; 6Arbovirology Department, Evandro Chagas Institute, Ananindeua 67030-000, Pará State, Brazil; 7Pathology Laboratory of Infectious Diseases (LIM50), Pathology Department, Medical School, São Paulo University, São Paulo 1246-903, São Paulo State, Brazil

**Keywords:** New and Old World, visceral leishmaniasis agents, *Leishmania (L.) donovani*, *Leishmania (L.) infantum*, *Leishmania (L.) infantum chagasi*, comparative genomic analyses

## Abstract

Visceral leishmaniasis (VL), also known as kala-azar, is an anthropozoonotic disease affecting human populations on five continents. Aetiologic agents belong to the *Leishmania (L.) donovani* complex. Until the 1990s, three leishmanine parasites comprised this complex: *L. (L.) donovani* Laveran & Mesnil 1903, *L. (L.) infantum* Nicolle 1908, and *L. (L.) chagasi* Lainson & Shaw 1987 (=*L. chagasi* Cunha & Chagas 1937). The VL causal agent in the New World (NW) was previously identified as *L. (L.) chagasi*. After the development of molecular characterization, however, comparisons between *L. (L.) chagasi* and *L. (L.) infantum* showed high similarity, and *L. (L.) chagasi* was then regarded as synonymous with *L. (L.) infantum*. It was, therefore, suggested that *L. (L.) chagasi* was not native to the NW but had been introduced from the Old World by Iberian colonizers. However, in light of ecological evidence from the NW parasite’s enzootic cycle involving a wild phlebotomine vector (*Lutzomyia longipalpis*) and a wild mammal reservoir (the fox, *Cerdocyon thous*), we have recently analyzed by molecular clock comparisons of the DNA polymerase alpha subunit gene the whole-genome sequence of *L. (L.) infantum chagasi* of the most prevalent clinical form, atypical dermal leishmaniasis (ADL), from Honduras (Central America) with that of the same parasite from Brazil (South America), as well as those of *L. (L.) donovani* (India) and *L. (L.) infantum* (Europe), which revealed that the Honduran parasite is older ancestry (382,800 *ya*) than the parasite from Brazil (143,300 *ya*), *L. (L.) donovani* (33,776 *ya*), or *L. (L.) infantum* (13,000 *ya*). In the present work, we have now amplified the genomic comparisons among these leishmanine parasites, exploring mainly the variations in the genome for each chromosome, and the number of genomic SNPs for each chromosome. Although the results of this new analysis have confirmed a high genomic similarity (~99%) among these parasites [except *L. (L.) donovani*], the Honduran parasite revealed a single structural variation on chromosome 17, and the highest frequency of genomic SNPs (more than twice the number seen in the Brazilian one), which together to its extraordinary ancestry (382,800 *ya*) represent strong evidence that *L. (L.) chagasi*/*L. (L.) infantum chagasi* is, in fact, native to the NW, and therefore with valid taxonomic status. Furthermore, the Honduran parasite, the most ancestral viscerotropic leishmanine parasite, showed genomic and clinical taxonomic characteristics compatible with a new *Leishmania* species causing ADL in Central America.

## 1. Introduction

Visceral leishmaniasis (VL) is an anthroponotic or zoonotic disease that affects human populations on five continents. According to the World Health Organization (WHO), VL causes approximately 50,000 to 90,000 new cases annually [1]. More than 90% of VL cases occur in six countries: India, Bangladesh, Sudan, South Sudan, Ethiopia, and Brazil [2]. In Latin America, it is known as American visceral leishmaniasis (AVL) or “neotropical calazar” [3,4].

AVL is a noncontagious infectious disease caused by a protozoan parasite of the order Kinetoplastida, family Trypanosomatidae, and genus *Leishmania* Ross 1903; the species of the aetiologic agent was known as *Leishmania (Leishmania) chagasi* Lainson & Shaw 1987 (=*Leishmania chagasi* Cunha & Chagas 1937) at the end of the last century [5]. Cunha & Chagas (1937) [6] described a new agent responsible for the disease in America, as it was distinct from the disease in Europe (the American parasite did not produce an experimental disease in domestic dogs, as did the European agent).

One of the most controversial aspects of AVL, however, concerns its true origin and the taxonomy of its agent. Some authors began to doubt the autochthonous nature of the disease agent near the end of the last century, as well as the origins of its infections. Momen et al. (1987) [7] first suggested that *Leishmania (L.) infantum* was the aetiologic agent of AVL based on comparisons of the enzyme profiles (11 enzyme loci) of 110 Brazilian isolates of *L. (L.) chagasi* from humans, dogs, and *Didelphis* sp. (opossum) with the WHO reference strain of *L. (L.) infantum*—with no differences observed among them. In that same year, Grimaldi et al. (1987) [8], using a set of 16 *Leishmania (L.) donovani* species-specific monoclonal antibodies, and Kreutzer et al. (1987) [9], working with 20 enzyme loci, were also unable to separate *L. (L.) chagasi* isolates from Brazil, Honduras, Panama, or Colombia from the WHO reference strain of *L. (L.) infantum*. Later, Momen et al. (1993) [10] once again presented evidence from enzyme electrophoresis and schizodeme analysis that *L. (L.) chagasi* had a recent origin in the New World and was similar to *L. (L.) infantum*.

On the other hand, and still, at the end of the 20th century, Mauricio et al. (1999) [11] pointed out the genomic diversity within the *L. (L.) donovani* complex and suggested that *L. (L.) chagasi* was synonymous with *L. (L.) infantum*. At the beginning of the present century, Maurício et al. (2000) [12] again addressed that issue using restriction fragment length polymorphism (RFLP) and random amplified polymorphic DNA (RAPD) methodologies, and the results reinforced their previous interpretation that *L. (L.) chagasi* was synonymous with *L. (L.) infantum*—thus suggesting that *L. (L.) chagasi* was introduced into the New World principally through Portuguese and/or Spanish colonization. Lukes et al. (2007) [13] later used a multifactorial genetic analysis (also based on RFLP and RAPD methodologies) that likewise showed *L. (L.) chagasi* to be indistinguishable from *L. (L.) infantum.* Those authors proposed that *L. (L.) donovani* and *L. (L.) infantum* should be the only recognized species of the *L. (L.) donovani* complex.

At the beginning of the last decade, Leblois et al. (2011) [14] addressed the origin of *L. (L.) chagasi*, its timing, and its demography, with fast-evolving genetic markers, a suite of Bayesian clustering algorithms, and coalescent modeling using 14 microsatellite markers from 450 strains from the *L. (L.) donovani* complex. The majority of the Central and South American *L. (L.) chagasi* were nested within the Portuguese *L. (L.) infantum* clade, indicating that AVL is one more disease that the “Conquistadores” carried to the New World. In that same year, Kuhls et al. (2011) [15] compared 98 *L. (L.) infantum* [=*L. (L.) chagasi*] isolates from different New World foci with 308 *L. (L.) infantum* and 20 *L. (L.) donovani* strains from Old World countries using the same multilocus microsatellite typing method. They concluded that *L. (L.) infantum* was the agent of AVL and that the parasite had been imported (recently and multiple times) to the NW from southwestern Europe. The same multilocus microsatellite typing method was used in Brazil to analyze the population structure of *L. (L.) infantum* [=*L. (L.) chagasi*] strains isolated from humans and dogs from most Brazilian states with endemic VL [16], as well as from dogs diagnosed with visceral leishmaniasis in São Paulo State [17]. Both studies showed the genetic structures of the parasites to be similar to those observed in the abovementioned works [14,15].

However, even though the above evidence shows that *L. (L.) chagasi* or *L. (L.) infantum chagasi* Lainson & Shaw 2005 would be synonymous with *L. (L.) infantum*, we recently reported, for the first time, the whole-genome sequence of *L. (L.) infantum chagasi* of the most prevalent clinical form, atypical dermal leishmaniasis (ADL), from Honduras (Central America). We compared it with that of the same parasite of the fox *Cerdocyon thous* from Brazil (South America), as well as with those of *L. (L.) donovani* (India), and *L. (L.) infantum* (Europe) by using molecular clock analysis of the DNA polymerase alpha subunit gene (a highly conserved genomic region related to the evolutionary process of leishmanine parasites). Our results revealed that *L. (L.) infantum chagasi* from Honduras is considerably more ancient (382,800 *ya*) than the same parasite from Brazil (143,300 *ya*), *L. (L.) donovani* (33,776 *ya*), and *L. (L.) infantum* (13,000 *ya*), which represents strong evidence that *L. (L.) chagasi*/*L. (L.) infantum chagasi* is, in fact, native to the New World (more precisely Central America) and not to the Old World [18].

Additionally, we have now amplified the genomic comparisons among these leishmanine parasites. We explored variations in the genome for each chromosome, and the number of genomic SNPs for each chromosome. Thus, although the results of this new analysis confirmed high genomic similarity (~99%) between these parasites [except *L. (L.) donovani*], genomic differences in the Honduran parasite should be highlighted. The Honduran parasite had a single structural variation on chromosome 17, and the highest frequency of genomic SNPs (more than twice the number seen in the Brazilian parasite), which reinforces its extraordinary ancestry (382,800 *ya*) in relation to the same parasite from Brazil (143,300 *ya*), to *L. (L.) donovani* (33,776 *ya*) from India and, finally, to *L. (L.) infantum* (13,000 *ya*) from Europe [18].

These findings, which represent the focus of this work, suggest the need for a taxonomic review of this group of parasites that have been allocated into the *L. (L.) donovani* complex. Such a taxonomic review is necessary because, while *L. (L.) infantum* was first described in relation to *L. (L.) infantum chagasi,* the exceptional ancestral character of the Honduran parasite is now evident, as well as its true geographic origin; likewise, the single structural variation on chromosome 17, the highest frequency of genomic SNPs (more than twice the number seen in the Brazilian parasite), and strong tropism for human skin all appear to be taxonomically consistent with the proposal that the original *L. (L.) donovani* complex must contain not only the three valid and chronologically described species [*L. (L.) donovani* Laveran & Mesnil 1903, *L. (L.) infantum* Nicolle 1908, and *L. (L.) chagasi* Cunha & Chagas 1937] but also a new *Leishmania* species representing the parasite that is responsible for the most prevalent clinical form of the disease [atypical cutaneous leishmaniasis (ACL) or nonulcerated cutaneous leishmaniasis (NUCL)] in Central American countries.

## 2. Materials and methods

### 2.1. Parasites

The two *Leishmania* spp. isolates used for this analysis are described below:-*L. (L.) infantum chagasi*: MCER/BR/1981/M6445/Salvaterra/Pará State/Brazil, isolated from the viscera (liver and spleen) of its wild reservoir, the crab-eating fox *Cerdocyon thous* (which appeared to be in good health) [19];-*L. (L.) infantum chagasi*: MHOM/HD/2017/M32502/Isla del Tigre/Amapala District/Honduras, isolated from a human case of atypical dermal leishmaniasis (ADL) or nonulcerated cutaneous leishmaniasis (NUCL) [20].

### 2.2. Parasite Cultivation and DNA Viability

Both *L. (L.) infantum chagasi* isolates were grown at 25 °C for seven days in Schneider medium supplemented with 10% fetal bovine serum, 10 μg/mL of 1% L-glutamine, and 100 IU/mL ampicillin; 3 mL aliquots were then collected for DNA extraction using the Relia Prep gDNA Mini Prep System (PROMEGA, Madison, WI, USA).

### 2.3. Parasite Sequencing Conditions

After *L. (L.) infantum chagasi* DNA extraction, the concentration used for sequencing was 1 ng/uL on the Illumina Hiseq (Illumina) platform. Total DNA quality was assessed using a NanoDrop 2000^TM^ spectrophotometer (ThermoFisher Scientific, Waltham, MA, USA). The genomic library was prepared using the Nextera XT DNA sample preparation kit (Illumina, San Diego, CA, USA). The quality of the library was verified using a Bioanalyzer 2010 (Agilent Technologies, Santa Clara, CA, USA), and the library was sequenced on a HiSeq 2500 instrument (Illumina, USA) with a 2× 100-bp paired-end format sequencing kit v.4. The DNA concentration used for sequencing was 1 ng/uL on the Illumina Hiseq (Illumina) platform.

### 2.4. Genome Analysis

The generated reads were trimmed with Trimmomatic v.0.39 [21] and assembled using a *de novo* strategy with SPAdes v software v.3.12 [22]. Genomes were manually curated and the final genomes were compared to those of *L. (L.) infantum* reference strains (FR796448 Spain, Europe, and CP027814 Turkey, Euro-Asia) using Geneious v.8.1.9 [23].

Genome annotation was performed with Augustus v.2.5.5 for *de novo* gene prediction software. To transfer the annotation from conserved sequences we used RATT (http://ratt.sourceforge.net/ (accessed on 12 January 2022) and *Leishmania (V.) braziliensis* (MHOM/BR/75/M2903) as a reference, while the annotation from dissimilar coding sequences was obtained using Pfam, as described in the COMPANION pipeline [24]. The genomes were aligned and analyzed for GC content, mean coverage, chromosome length, and nucleotide identity with Geneious v.8.1.9 [23]. The genome information was plotted using a Seaborn package [25].

### 2.5. Ethical Approval

This project was approved by the Research Ethics Committee of the Medicine School of the University of São Paulo (CAAE protocol: 64223917.1.0000.0065).

## 3. Results

The principle genomic comparisons among *L. (L.) infantum chagasi* from Brazil (South America) and Honduras (Central America) and the European reference strain *L. (L.) infantum* (JPCM5) are shown in Table 1.

The total lengths of both genomes were 31,924,889 nt, with 36 chromosomes that varied in length, as shown in Figure 1; the total number of genes predicted for each genome was 8423. After the sequencing procedures, 36 chromosomes were initially identified for the two strains, with length profiles shown in Figure 1. A progressive increase in chromosome lengths from 1 to 36 can be observed.

The genomes of *L. (L.) infantum* from Europe and *L. (L.) infantum chagasi* from Honduras/Brazil demonstrated approximately 99% identity and varied mainly in chromosome 17, which had a single structural variation. Compared to the other *Leishmania* species, the variation in chromosome 17 was greater in the Honduran parasite, as shown in Figure 2.

The genomic comparisons among *L. (L.) infantum chagasi* from Brazil (South America) and Honduras (Central America) and *L. (L.) infantum* (Europe) were also evaluated based on the numbers of SNPs in each genome—a genetic character linked to the ancestry of the parasite. We observed that the Honduran parasite had more than twice as many SNPS (44,627) as the Brazilian parasite (16,867), demonstrating its older ancestry among the leishmanine parasites examined (Figure 3).

## 4. Discussion

Our results represent the first genomic analysis performed in Latin America on whole-genome sequences of *L. (L.) infantum chagasi* from Honduras (Central America), the same parasite from Brazil (South America), and sequences of *L. (L.) infantum* (Europe) and *L. (L.) donovani* (India). The three parasite genomes revealed high genomic identity (~99%) [except *L. (L.) donovani*], although previously molecular clock comparisons of the DNA polymerase alpha subunit gene (a highly conserved genomic region related to the evolutionary process of leishmanine parasites [26]) among these parasites indicated that the Honduran parasite was significantly older (382,800 *ya*) than the Brazilian parasite (143,300 *ya*), *L. (L.) donovani* (33,776 *ya*), or *L. (L.) infantum* (13,000 *ya*) [18]. Moreover, the Honduran parasite was found to have a single structural variation (and perhaps also a single functional variation) on chromosome 17 (Figure 2), as well as the highest frequency of genomic SNPs (more than twice that of the Brazilian parasite) (Figure 3). The original data presented here contradict recent views concerning the origin of the *Leishmania* parasite responsible for AVL as well as its taxonomic position [12,13,14,15,27].

These results also appear to contradict the interpretation that the low genomic heterogeneity of *L. (L.) infantum chagasi* samples from Latin America (mostly from Brazil) is due to the recent introduction of that parasite into the New World [15]. Our results clearly suggest that the low genomic heterogeneity of *L. (L.) infantum chagasi* reflects the parasite’s low genomic plasticity (due to its older ancestry) much more than its recent introduction into the New World. Likewise, another species belonging to the *L.* (*Leishmania*) subgenus [*L. (L.) amazonensis*], an ancient species of this subgenus that was identified in previous work [18]), evidenced low genomic plasticity in relation to other species of later ancestry in the subgenus *L. (Viannia)* [such as *L. (V.) braziliensis* and *L. (V.) panamensis*], suggesting that low plasticity may be a genomic characteristic of the subgenus *L.* (*Leishmania*) [28,29,30,31].

Indeed, the high genomic identity (~99%) observed among these parasites [except *L. (L.) donovani*] has been crucial to the view that *L. (L.) infantum chagasi* is synonymous with *L. (L.) infantum* and that the causal agent of AVL was introduced to Latin America during Iberian colonization over the last 500 years [12,13,14,15,27]. However, it is interesting to note how difficult it is to explain the establishment of the parasite in Latin America in an ecological niche quite different from that of southwestern Europe. The Latin American niche includes a sandfly vector of a different genus (*Lutzomyia*) from that found in Europe (*Phlebotomus*) and wild mammalian hosts of different orders (rodent, marsupial, canid, feline, and chiroptera) [3,32,33,34,35,36,37,38,39,40]. Adaptation of the parasite to the main phlebotomine vector of AVL in Latin America, *Lutzomyia longipalpis*, has been suggested [32]. In doing so, previous authors have omitted some principles of ecology of the sandfly, which lives in a natural environment in Latin America that is alongside the greatest known global genetic diversity of *Leishmania* parasites. It is therefore relevant to note that the Latin American geographic region currently hosts representatives of the three known subgenera of the genus *Leishmania* (including *Leshmania*, *Viannia,* and *Mundinia*) [41], with a total of fifteen different well-known causal agents of American cutaneous leishmaniasis (ACL). In addition, Latin America hosts four hybrid parasites: *L. (V.) panamensis/L. (V.) braziliensis*, *L. (V.) panamensis/L. (V.) guyanensis*, *L. (V.) braziliensis/L. (V.) peruviana*, and *L. (V.) guyanensis/L. (V.) shawi* as well as one causal agent of AVL—even without considering other *Leishmania* species that do not commonly infect humans, such as *L. (L.) enriettii*, *L. (L.) hertigi*, *L. (L.) deanei*, and *L. (L.) aristidesi* [3,37,42,43,44,45,46].

Thus, it does not seem likely that *Lutzomyia longipalpis*, a wild phlebotomine species considered permissive at the laboratory level [47,48,49], would no longer be involved in the transmission of any of those various *Leishmania* species in Latin America. Occasionally, there is evidence of *L. (V.) braziliensis* or *L. (L.) amazonensis* DNA in *Lutzomyia longipalpis* caught in the wild [50,51,52,53]—although these reports are not sufficient to incriminate this phlebotomine as a vector of leishmanine parasites. It is extremely relevant that the laboratory-bred sandfly vector can become experimentally infected after feeding on blood from the skin lesion edges of a patient with ACL caused by *L. (V.) braziliensis* [54]—reinforcing its potential as a vector for transmitting a *Leishmania* species other than *L. (L.) infantum chagasi*—which has never been definitively demonstrated under natural conditions. It, therefore, appears easier to accept that *Lutzomyia longipalpis* has developed a species-specific interaction with native *L. (L.) infantum chagasi* and its main wild reservoir (the crab-eating fox, *Cerdocyon thous*) over thousands of years and represents an ecologically balanced enzootic cycle that is widely distributed in Latin America [3,4,37,55,56,57]. It is not imperative that major genomic differences exist between *L. (L.) infantum chagasi* and *L. (L.) infantum* for this ecologically balanced enzootic cycle to occur in the New World, but only that *L. (L.) infantum chagasi* existed prior to the description of *L. (L.) infantum*—which seems clear based on the results of this work.

Corroborating the idea of a native enzootic cycle of the parasite in the New World is recent evidence pointing to the role of a small wild rodent (*Proechimys* sp.) in the enzootic cycle of *L. (L.) infantum chagasi* in a mountainous area of primary forest in the “Serra dos Carajás” (municipality of Marabá) in Pará State in the Brazilian Amazon [33]. European/African descendants only entered that area approximately 50 years ago—an extremely short window of time for the parasite to have adapted to wild rodents after the arrival of supposedly infected dogs.

More recent evidence has corroborated the wild enzootic cycle of the parasite in the Brazilian Amazon in the central area of Amapá State in the Wajãpi Indigenous Territory (WIT). The WIT area is a rugged landscape covered by dense rainforest and is part of one of the world’s largest continuous protected areas of rainforest, covering over 12 million hectares. A natural *L. (L.) infantum (chagasi)* infection was found there in two wild rodents (*Dasyprocta* sp. and *Proechimys cuvieri*) by PCR-based amplification of a fragment of the ITS1 gene (300–350 bp) 7-8 and by semi-nested PCR for a fragment of hsp70 (640 bp) [34]. Taken together, this information indicates the existence of a native enzootic cycle of *L. (L.) infantum chagasi* in the Brazilian Amazon, with the participation of not only a wild canid (*Cerdocyon thous*) but also small wild rodents (*Proechimys* sp., *Proechimys cuvieri,* and *Dasyprocta* sp.). Coincidence or not, it was precisely in this region that the original description of the AVL causal agent was made (*Leishmania chagasi* Cunha & Chagas 1937).

From the genomic point of view, the present results undoubtedly confirm a high similarity (~99%) among the studied parasites [except *L. (L.) donovani*] and, by extension, even isolated samples of asymptomatic human infection [12,13,14,15,27,58]. However, it is necessary to emphasize the presence of a single structural variation (and perhaps a functional variation) on chromosome 17 of the Honduran parasite in relation to the same parasite from Brazil, *L. (L.) infantum* reference strain [JPCM5] [59], and *L. (L) donovani* (Figure 2), although the full meaning of this variation has not yet been deciphered. Interestingly, there had been no reported AVL cases for at least ten years in the area (Isla del Tigre, Amapala District, West Coast of Honduras) when the parasite was isolated from a human case of atypical dermal leishmaniasis [ADL] (all symptomatic cases of human infection registered during that time were the ADL form). Whether there is any association between that single structural variation on chromosome 17 of the Honduran parasite and the biology of human infection is open to further investigation, although we speculate that an ecological process of host-parasite stabilization during the human-parasite ancestry process in Central American countries could have occurred. The situation in Costa Rica is an example; only the atypical cutaneous form of infection has been recorded in recent years (although a single case of AVL has now been diagnosed) [60].

Another possibility analogous to that of the Honduran parasite involves the RagC gene found on chromosome 36 of *L. (L.) donovani*, which controls GTPase protein expression and appears to be involved in the process of parasite visceralization in Sri Lanka in Asia [61,62]. It will therefore be necessary to screen the main genes on chromosome 17 of the Honduran parasite to search for a gene that might inhibit the parasite’s visceralization, as there is strong clinical evidence that human infections are evolving towards dermal stabilization in the area of the parasite’s origin [63,64].

Another original finding in this work is the high frequency of genomic SNPs in the Honduran parasite (more than twice the number found in the Brazilian parasite) (Figure 3), a fact that seems to be strongly related to the older ancestry of the Honduran parasite compared to the other parasites [18]. SNPs are known to have low mutation rates, which makes them useful genetic markers for following the inheritance patterns of chromosomal regions from generation to generation and excellent markers of ancestry [65].

There are currently two clinical forms (ADL and AVL) of human symptomatic *L. (L.) infantum chagasi* infections along the eastern coast of Honduras, with ADL being the most common. Previous research (using kDNA RFLP and RAPD techniques) has indicated that the two clinical forms (ADL and AVL) are caused by genetically similar parasites [66]. Additional observed genomic similarities (CG content percentages and mean coverage) supported a high genomic identity (~99%) among the parasites studied (Table 1) and confirmed that they constitute a genomically cohesive group. Although the results of the present genomic analysis corroborate previous works [12,13,14,15,58], the major question in this study concerns not only genomic differences among the parasites but their geographical origins.

As such, the complement to comparative genomic analyses of the parasites is molecular clock comparisons of the DNA polymerase alpha subunit gene (a highly conserved genomic region related to the evolutionary process of leishmanine parasites) [26]. Molecular clock comparisons have revealed that the Honduran parasite is significantly older (382,800 *ya*) than *L. (L.) infantum chagasi* from Brazil (South America), *L. (L.) donovani* (India), and *L. (L.) infantum* (Europe) [18]. These molecular clock data represent strong evidence that *L. (L.) infantum chagasi* is native to the New World (more precisely, Central America) and not to the Old World, as has been suggested [12,13,14,15,27]. One might counter this evidence by noting that the genomic analysis was based on only one strain of *L. (L.) infantum chagasi* from Honduras and Brazil, however, similar results have been obtained by PCR analyses examining three different genetic targets (12S, 9S, and ND7 of maxi circle genes) from a larger sample of parasites (4) showing the same clinical form of the disease (ADL) as well as the same geographic origin (Isla del Tigre, Amapala District, West Coast of Honduras). These results corroborate our finding that the Honduran parasite is a different lineage than the Brazilian parasite (Fernández Figueroa, unpublished data). Moreover, *Leishmania (Viannia) utingensis* Braga et al. 2003, a parasite that was described based on the phenotypic characteristics of a single isolate obtained from the sandfly *Lutzomyia (Viannamyia) tuberculata*, remains a valid species without any other isolate of the parasite from the same sandfly, from another wild mammal, or even from humans in the Brazilian Amazon. In a punctual analysis such as this, with a focus on the origin and/or the taxonomy of these parasites, the evaluation of only one individual of each species provided highly significant results for the origin (ancestry) of these parasites, and a broad genomic spectrum analysis of different endemic regions worldwide would have led to the 31 Latin American samples being treated as *L. (L.) infantum*, therefore neglecting *L. (L.) infantum chagasi* [67].

Another point that deserves to be highlighted refers to the accuracy of our previous molecular clock comparisons [18]. Although we used only the DNA polymerase alpha subunit gene to assess ancestry, we were able to identify a phylogenetic tree consisting of the two major clades within the genus *Leishmania* represented by the two subgenera with the greatest clinical interest, *L. (Leishmania)* and *L. (Viannia)*, and by the main leishmanine parasite species that act as aetiologic agents of human leishmaniases [including *L. (L.) donovani*, *L. (L.) infantum*, *L. (L.) infantum chagasi*, *L. (L.) aethiopica*, *L. (L.) major*, *L. (L.) mexicana* and *L. (L.) amazonensis*, and *L. (V.) braziliensis*, *L. (V.) peruviana*, *L. (V.) panamensis*, *L. (V.) guyanensis* and *L.(V.) lainsoni*]. Regarding the target of greatest interest in that work [i.e., the *L. (L.) donovani* complex, whose origin node was estimated at 1.4 *Mya*], it is important that our molecular clock comparisons using only the DNA polymerase alpha subunit gene were also able to identify four different nodes resulting from the evolutionary process of 1.4 *Mya*, which, in chronological order, corresponding to the following: (1) *L. (L.) infantum chagasi*—Honduras, Central America [382,000 *ya*]; (2) *L. (L.) infantum chagasi*—Brazil, South America [143,000 *ya*]; (3) *L. (L.) donovani*—India [33,776 *ya*]; and (4) *L. (L.) infantum*—Europe [13,000 *ya*]. Thus, we note that our phylogenetic design, based on DNA polymerase alpha subunit gene sequences, was able to reconstruct a phylogenetic tree with a podological profile substantially similar to trees reconstructed using other gene regions [29,68], demonstrating its application for phylogenetic studies of Leishmaniinae and related parasites. It is also important to mention the pioneering work that produced exceptionally high-value DNA and RNA polymerase gene sequences in phylogenetic studies of the genus *Leishmania.* Those studies considered several representative species of parasites from the *L. (Leishmania)* and *L. (Viannia)* subgenera [*L. (L.) tropica*, *L. (L.) donovani*, *L. (L.) mexicana*, *L. (L.) hertigi*, and *L. (V.) braziliensis*], as well as parasites of previously questionable taxonomy [*Leishmania herreri*, *Sauroleishmania adleri*, *Sauroleishmania deanei*, *Sauroleishmania gymnodactyli,* and *Sauroleishmania tarentolae*], which led to the proposal that *Leishmania* spp-infecting reptiles evolved from mammalian *Leishmania* spp. [69], a hypothesis that is now well-established in taxonomic classifications of the genus *Leishmania* [41,70].

Based on these results, there is now a clear need to determine not only the origins and historical geographical routes of dispersal of leishmanine parasites associated with visceral leishmaniasis throughout the world but also their taxonomic positions. This will be a difficult task, however, as the origins of these parasites date to approximately 90–100 million years ago [68]. The first significant step in this direction is the suggestion that “the predecessors of the *Leishmania (L.) donovani* group and *Leishmania (L.) major* would have evolved from monoxenous parasites of insects in South America (46–36 *Mya*) and moved to Asia via the Bering land bridge. The ancestor of the *L. (L.) donovani* complex diverged from other *Leishmania* sp. (14–24 *Mya*), arrived in central Asia, and diverged into European *L. (L.) infantum*, African, and Indian/Kenyan *L. (L.) donovani*”. This ancestral line makes the case for this group of parasites originating in the New World (more precisely in South America) [13]. Thus, it is not surprising that the results of our previous work show the likely origin of *L. (L.) infantum chagasi* as the New World. However, unlike this previous proposal, *L. (L.) infantum chagasi* is considered to have a more specific origin in the Central American region [18].

It, therefore, seems reasonable to assume that the parasite migrated from the Central American region to Asia, following the itinerary suggested above, and then diverged into European and North African [*L. (L.) infantum*], and into African and Indian/Kenyan [*L. (L.) donovani*] [13]. The geographic dispersal of the parasite may also have been directed southwards by the migration of wolves of approximately 10 *Mya* from North America to Central America, finally arriving in South America [71]. There is also a strong ecological relationship between the parasite [*L. (L.) infantum chagasi*] and a widely distributed wild canid species in Latin America, the crab-eating fox *Cerdocyon thous* [19], which is a strong attraction for the main sandfly vector of the parasite, *Lutzomyia longipalpis* [32]. Reinforcing this hypothesis of a North American to South American axis, a similar geographic trajectory may also have been made by the North American *L. (L.) mexicana*, giving rise to *L. (L.) pifanoi* and *L. (L.) amazonensis* in South America. Strong evidence in that regard is records of ACL caused by *L. (L.) mexicana* and *L. (L.) amazonensis* in Ecuador, suggesting the possibility of a hybridization zone between those two parasites [72].

There is a real need for a taxonomic review of this group of parasites, hitherto allocated into the *L. (L.) donovani* complex. While *L. (L.) infantum* was described first, it is now evident that the species was derived through a long process of speciation from the ancestral Central American parasite, known until then as *L. (L.) infantum chagasi*. Thus, considering the exceptional ancestral character (382,800 *ya*) of the Honduran parasite (Central America) that gave rise to the speciation process of the leishmanine parasites currently associated with the aetiology of visceral leishmaniasis worldwide [i.e., *L. (L.) donovani*, *L. (L.) infantum* and *L. (L.) infantum chagasi*], it seems taxonomically consistent that the original *L. (L.) donovani* complex must contain not only the three valid species already described [*L. (L.) donovani* Laveran & Mesnil 1903 (Africa/India), *L. (L.) infantum* Nicolle 1908 (Europe/North Africa), and *L. (L.) chagasi* Cunha & Chagas 1937 (South America)] but also, exceptionally, a new *Leishmania* species representing the leishmanine parasite in the Central American countries that is strongly associated with atypical cutaneous leishmaniasis (ACL) or nonulcerated cutaneous leishmaniasis (LCNU). That putative species seems to be signaling a strong biological character of this parasite (a strong tropism for human skin), which is distinct from descendants that demonstrated typical visceral tropisms. In line with this proposal, it is relevant that the taxonomic classification of the genus *Leishmania* is not only well-defined phylogenetically but can also be described in terms of host specificity (vertebrate and/or invertebrate) or by clinical parameters [70]. Thus, the need to define the specific identity of the Central American (Honduras) ancestral parasite seems clear and will be addressed in the near future.

## Figures and Tables

**Figure 1 microorganisms-11-00025-f001:**
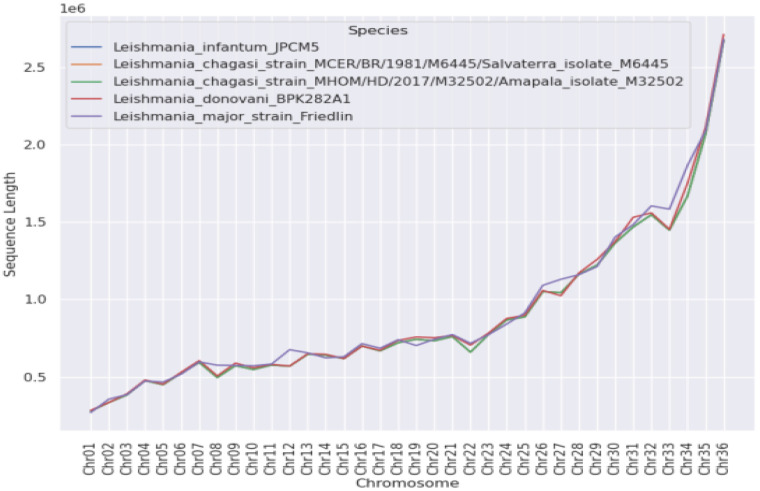
Lengths (Mbp) of the individual chromosomes identified and their similarities to other *Leishmania* species: *L. (L.) infantum chagasi* (Brazil); *L. (L.) infantum chagasi* (Honduras); *L. (L.) donovani*; *L. (L.) infantum* (Europe); and *L. (L.) major*.

**Figure 2 microorganisms-11-00025-f002:**
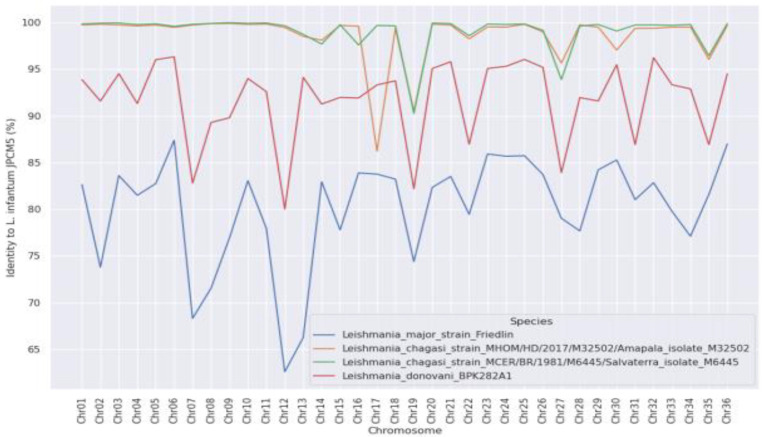
Variations in genome identity for each chromosome and in other *Leishmania* species; note a single structural variation of chromosome 17 of the Honduran parasite compared to the other *Leishmania* species: *L. (L.) infantum chagasi* (Brazil); *L. (L.) infantum chagasi* (Honduras); *L. (L.) donovani*; and *L. (L.) major*.

**Figure 3 microorganisms-11-00025-f003:**
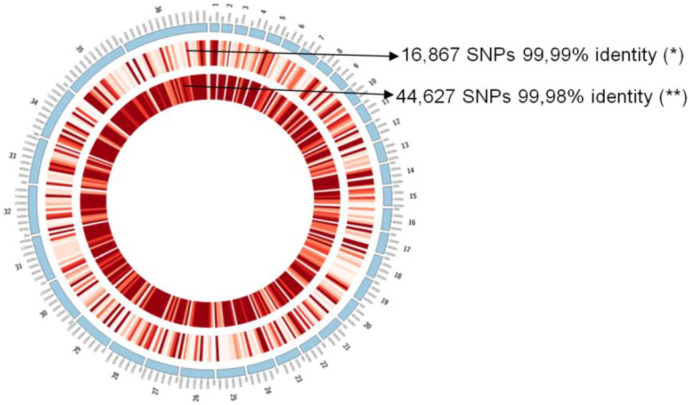
Genomic comparisons among *L. (L.) infantum chagasi* from Brazil (*) and Honduras (**), and *L. (L.) infantum* from Europe (reference strain JPCM5). The bars represent 100 kb, and the red gradient represents the number of SNPs found in each region; the deeper the red color, the greater the number of SNPs found.

**Table 1 microorganisms-11-00025-t001:** *De novo* genomic assembly of *L. (L.) infantum chagasi* from Brazil/Honduras and comparison with *L. (L.) infantum* (Europe).

Sample Code	No. Reads #/ Contigs †	Min. Size † (bp)	Max. Size † (bp)	*N*_50_ † (bp)	Coverage †	GC Content † (%)	Identity (%) †
MCER/BR/1981/M6445 *	68,908,876 15,315	200	60,666	8354	130×	59.6	99.99%
MHOM/HD/2017/M32502 **	25,730,242 7193	200	57,312	8724	53.47×	59.3	99.98%

* *L. (L.) infantum chagasi* from Brazil/** *L. (L.) infantum chagasi* from Honduras. † Previously published data [18]. **#** Unpublished data.

## Data Availability

The data that support the findings of this study are available from the corresponding author upon reasonable request.

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
