# Peer review of "Comparative Genomic Analyses of New and Old World Viscerotropic Leishmanine Parasites: Further Insights into the Origins of Visceral Leishmaniasis Agents"

_microorganisms, 2022, doi:10.3390/microorganisms11010025_

Round 1

Reviewer 1 Report

Dear Authors, I felt like I was reading a draft of your manuscript. Several typos are present. Many sentences must be rephrased, given that they are not clear.  Information relating to the equipment must be better detailed. 

Visceral leishmaniasis, also known as kala-azar, is an anthropozoonotic disease transmitted through the bites of infected female phlebotomine sandflies, which feed on blood to produce eggs. The disease is diffused on five continents and is caused by a protozoa parasite - Leishmania (L.) donovani complex including  L. (L.) donovani, L. (L.) infantum, and L. (L.) chagasi. With the development of molecular characterization, it has been reported a high similarity between L. (L.) chagasi and L. (L.) infantum,  evidencing L. (L.) chagasi as synonymous with L. (L.) infantum. Given that the New World parasite’s enzootic cycle involving phlebotomine vector (Lutzomyia longipalpis) and reservoir (the fox, Cerdocyon thous) wild, the study we report here the first molecular analysis of the whole-genome sequence of L. (L.) infantum chagasi) of the most prevalent clinical form (atypical dermal leishmaniasis) from Honduras (Central America), comparing it with the form of L. (L.) infantum and L. (L.) donovani isolated in fox from Brazil (South America). In light of ecological evidence of involvement of new phlebotomine vector (Lutzomyia longipalpis) and wild reservoir (the fox, Cerdocyon thous), the topic may be considered original, adding  new information regarding the taxonomy of Leishmania in Latin-America. The MS is well organized an structured, conclusion are consistent with the aim of the study and appropriate references are included.

Author Response

Microorganisms - Reviewer 1.

Dear Authors, I felt like I was reading a draft of your manuscript. Several typos are present. Many sentences must be rephrased, given that they are not clear.  Information relating to the equipment must be better detailed. 

Visceral leishmaniasis, also known as kala-azar, is an anthropozoonotic disease transmitted through the bites of infected female phlebotomine sandflies, which feed on blood to produce eggs. The disease is diffused on five continents and is caused by a protozoa parasite - Leishmania (L.) donovani complex including  L. (L.) donovani, L. (L.) infantum, and L. (L.) chagasi. With the development of molecular characterization, it has been reported a high similarity between L. (L.) chagasi and L. (L.) infantum,  evidencing L. (L.) chagasi as synonymous with L. (L.) infantum. Given that the New World parasite’s enzootic cycle involving phlebotomine vector (Lutzomyia longipalpis) and reservoir (the fox, Cerdocyon thous) wild, the study we report here the first molecular analysis of the whole-genome sequence of L. (L.) infantum chagasi) of the most prevalent clinical form (atypical dermal leishmaniasis) from Honduras (Central America), comparing it with the form of L. (L.) infantum and L. (L.) donovani isolated in fox from Brazil (South America). In light of ecological evidence of involvement of new phlebotomine vector (Lutzomyia longipalpis) and wild reservoir (the fox, Cerdocyon thous), the topic may be considered original, adding  new information regarding the taxonomy of Leishmania in Latin-America. The MS is well organized an structured, conclusion are consistent with the aim of the study and appropriate references are included.

First of all, let us thank you for your work in reviewing our manuscript submitted to Microorganisms. 

We would like to inform you that the manuscript has been fully revised seeking to meet not only your suggestions and recommendations, but also those of other reviewers.We hope we have met your comments and requests.

Reviewer 2 Report

The study was well designed, results were good presented but the manuscript have many grammatical mistakes, it must be edit by native speaker

Author Response

Microorganisms - Reviewer 2.

The study was well designed, results were good presented but the manuscript have many grammatical mistakes, it must be edit by native speaker.

First of all, let us thank you for your work in reviewing our manuscript submitted to Microorganisms. 

We would like to inform you that the manuscript has been fully revised seeking to meet not only your suggestions and recommendations, but also those of other reviewers.We hope we have met your comments and requests.

Reviewer 3 Report

The work describes the Comparative genomic analyzes of New and Old World viscerotropic leishmanine parasites: Further insights into the origins of visceral leishmaniasis agents. Unfortunately, in this manuscript's current form I cannot fully assess the results presented due to errors and issues with the structuring of the work. It's of note that a read through the manuscript to correct formatting, grammar, and spelling should be done as numerous issues were found throughout the manuscript.

- The abstract should include the following elements: place the introduction addressed in a broad context, highlight the purpose of the study, briefly describe the main methods applied, summarize the main findings, and indicate the main conclusions.

- Keyword: the integration of the terms of the title must be avoided

- Introduction: What is the problematic and what is the added value of this work?

- Authors are encouraged to use line numbering to facilitate revision

-Latin names of the species should be written in italics

- “Materials and methods” instead of “Materials and methods”

- Chart 1: "." instead "," (for example “60.666” instead “60,666”

- Redraw figures 1 and 2

- Picture 3 "." instead ","

- Authors are invited to respect the instructions of the journal

Author Response

Microorganims - Reviewer 3.

The work describes the Comparative genomic analyzes of New and Old World viscerotropic leishmanine parasites: Further insights into the origins of visceral leishmaniasis agents. Unfortunately, in this manuscript's current form I cannot fully assess the results presented due to errors and issues with the structuring of the work. It's of note that a read through the manuscript to correct formatting, grammar, and spelling should be done as numerous issues were found throughout the manuscript.

First of all, let us thank you for your work in reviewing our manuscript submitted to Microorganisms. 

We would like to inform you that the manuscript has been fully revised seeking to meet not only your suggestions and recommendations, but also those of other reviewers.

- The abstract should include the following elements: place the introduction addressed in a broad context, highlight the purpose of the study, briefly describe the main methods applied, summarize the main findings, and indicate the main conclusions.

We revised the abstract according to your suggestions, hoping to have met your expectations.

- Keyword: the integration of the terms of the title must be avoided

We have already implemented the suggested change in the text of the manuscript.

- Introduction: What is the problematic and what is the added value of this work?

As presented in the Introduction of the manuscript, the problem or the essence of the work refers to the true origin of the parasite, the Leishmania species which is the agent of visceral leishmaniasis in Latin America. Historically, the parasite was described in 1937 by Cunha & Chagas as Leishmania chagasi, based on its biological behavior in experimental infection in domestic dogs, which remained valid as a species until the end of the last century. Already in the first two decades of this century, molecular studies comparing the causative agent of the disease in the New World, Leishmania chagasi, with the species causing the same disease in the Old World, Leishmania infantum, have revealed great similarity between the two species and, based on dating the description of both, as Leishmania infantum was described first, in 1908, while Leishmania chagasi in 1937, the species Leishmania chagasi has been disregarded. However, based on very solid evidence about the ecology of Leishmania chagasi in the Brazilian Amazon, which is totally different from that of Leishmania infantum in the Old World, we proposed the present work by comparing the genome of these parasites, Leishmania chagasi from the Brazilian Amazon (South America ) and Leishmania infantum from Spain (Europe), as well as Leishmania chagasi from Honduras (Central America) and Leishmania donovani from India (), which revealed through the molecular clock technique that Leishmania chagasi from Honduras and the Brazilian Amazon are significantly more ancestral that Leishmania donovani and Leishmania infantum, showing that Leishmania chagasi (=Leishmania infantum chagasi) is native to the New World and, therefore, represent a valid parasitological entity (species).

- Authors are encouraged to use line numbering to facilitate revision

Already answered in the revised version of the manuscript.

-Latin names of the species should be written in italics

Already answered in the revised version of the manuscript.

- “Materials and methods” instead of “Materials and methods”

Already answered in the revised version of the manuscript.

- Chart 1: "." instead "," (for example “60.666” instead “60,666”

Already answered in the revised version of the manuscript.- Redraw figures 1 and 2 Already answered in the revised version of the manuscript.

- Picture 3 "." instead ","

Already answered in the revised version of the manuscript.

- Authors are invited to respect the instructions of the journal.

Allow us to explain that, recently (last October), we published another work in Microorganisms journal and we formatted the present one based on the previous one, but, unfortunately, at the time we submitted the current file (in word) it is possible that some changes have occurred. We apologize and inform you that we have already made the corrections.We hope we have met your comments and requests.

Round 2

Reviewer 3 Report

The manuscript has been improved.